# Leveraging synthetic data produced from museum specimens to train adaptable species classification models

**Jarrett D. Blair**[1,2]*, **Kamal Khidas**[3,4], **Katie E. Marshall**[1]

1 Department of Zoology, University of British Columbia, Vancouver, British Columbia, Canada,
2 Department of Ecoscience, Aarhus University, Aarhus, Denmark, 3 Beaty Center for Species Discovery, Canadian Museum of Nature, Ottawa, Ontario, Canada, 4 Biology Department, Laurentian University, Sudbury, Ontario, Canada

* jarrett@ecos.au.dk

**Data availability statement:** All data files are are available on Zenodo (https://doi.org/10.5281/zenodo.15038538). All code is available in a public GitHub repository (https://doi.org/10.5281/zenodo.15536489).

## Abstract

Computer vision has increasingly shown potential to improve data processing efficiency in ecological research. However, training computer vision models requires large amounts of high-quality, annotated training data. This poses a significant challenge for researchers looking to create bespoke computer vision models, as substantial human resources and biological replicates are often needed to adequately train these models. Synthetic images have been proposed as a potential solution for generating large training datasets, but models trained with synthetic images often have poor generalization to real photographs. Here we present a modular pipeline for training generalizable classification models using synthetic images. Our pipeline includes 3D asset creation with the use of 3D scanners, synthetic image generation with open-source computer graphic software, and domain adaptive classification model training. We demonstrate our pipeline by applying it to skulls of 16 mammal species in the order Carnivora. We explore several domain adaptation techniques, including maximum mean discrepancy (MMD) loss, fine-tuning, and data supplementation. Using our pipeline, we were able to improve classification accuracy on real photographs from 55.4% to a maximum of 95.1%. We also conducted qualitative analysis with t-distributed stochastic neighbor embedding (t-SNE) and gradient-weighted class activation mapping (Grad-CAM) to compare different domain adaptation techniques. Our results demonstrate the feasibility of using synthetic images for ecological computer vision and highlight the potential of museum specimens and 3D assets for scalable, generalizable model training.

## Introduction

The field of ecology is transitioning into a 'big data' science [1–3]. This means that the volume and velocity of data required for modern ecological analytics is surpassing the capacity of individual researcher's ability to collect and process [1,4]. As such, manual data collection methods such as visual taxonomic classification have become a bottleneck in the ecological data acquisition pipeline [5]. To alleviate this bottleneck, significant effort has

**Funding:** JDB received a Mitacs Accelerate grant to fund this work. The grant does not have a specific grant number. Mitacs' website can be found at https://www.mitacs.ca/. The funders did not play any role in the study design, data collection, decision to publish, or preparation of the manuscript.

**Competing interests:** The authors have declared that no competing interests exist.

gone into automating the classification process through tools such as computer vision [6–8]. However, building effective computer vision tools for ecological research presents its own unique challenges. Among the most pervasive of these challenges is collecting high quality, annotated data to train the classification algorithms used by these tools [9,10]. This challenge stems from two main underlying issues: a lack of human resources to collect and process data, and, in some cases, few biological replicates. Applications such as iNaturalist and Wildlife Insights have addressed the human resource issue through massive crowd-sourcing and volunteer campaigns to collect their images and annotations, but such campaigns present their own logistical challenges [11,12]. Additionally, crowdsourced data often has a taxonomic bias towards charismatic and conspicuous fauna, and thus the issue of a lack of biological replicates for the majority of species persists [13].

Oversampling via image augmentation is a potential solution to both the human resource and biological replicate challenges of collecting high quality image datasets. Generally, oversampling refers to the process of generating additional training images from existing data, and image augmentation refers to the application of transformations (cropping, rotations, colour adjustments, etc.) to images [9,14]. This can greatly increase the size of a training dataset without any additional specimen collection or annotation effort. While many forms of image augmentation are applied through post-processing raw images, other forms of image augmentation work by changing how the images are originally captured. One example is multiview augmentation, which works by capturing several images of a single specimen from multiple angles or postures [15–17]. While 2D images are limited to a single perspective, multiview augmentation works in three dimensions, which enhances the model's feature learning and generalization better than standard augmentations made to raw 2D images. This is because the same specimen, when viewed from multiple perspectives (dorsal, ventral, side, anterior, etc.), can appear very differently on a two-dimensional plane. If a model never sees a given perspective during training, it might not be able to generalize its knowledge to that perspective when the model is deployed. However, multiview augmentation can often have the disadvantage of requiring greater manual sampling effort compared to other oversampling techniques as it requires multiple views of the same specimen [18,19].

Another oversampling technique is the use of "synthetic images" such as rendered images of 3D assets. 3D assets can be generated from real specimens by using methods such as photogrammetry, or completely synthetically through artistic creation [20,21]. Similar to multiview augmentation, 3D assets can be rendered from different perspectives, in different postures, or under different imaging conditions [19,22]. However, unlike with manual multiview augmentation, this can all be done automatically and repeatedly once the 3D asset is created. Additionally, 3D assets allow for otherwise-destructive augmentations to be made (e.g., removing legs from a beetle mesh) without damaging the original specimen. Given that all these techniques (destructive, multiview, and other image augmentations) can all be applied in combination automatically, 3D assets have considerable potential to generate large amounts of synthetic images from a relatively small number of specimens.

The motivation for creating 3D assets from real specimens goes beyond their applicability in computer vision. To improve the accessibility of their specimens, museums have gone to great effort to digitise their natural history collections, primarily through the use of 2D imaging [23–26]. However, by definition, 2D images contain far less information than 3D assets [27–29]. Even when they are used for 1-dimensional morphometric measurements, 2D images are less robust than 3D assets due to parallax errors (i.e., errors due to changes in viewing angles) [21]. Therefore, by capturing anatomically-correct 3D representations of museum specimens, we can simultaneously improve museum collection digitization while also creating a valuable resource for computer vision model training.

Although synthetic images can be used to build large image datasets, most computer vision models in ecology are used to classify real photographs. This presents a significant challenge to using this type of workflow as a source of training data because synthetic images belong to a different image domain (i.e., image type) than images taken using standard photography [30–32]. While synthetic images produced from renders of 3D assets often appear photorealistic, computer vision models may still pick up on subtle differences, thus making model generalization across domains more difficult [33–36]. To date, most research on synthetic animal images in computer vision has focussed on pose estimation [22,31,37–40]. When used for classification, challenges in domain generalization have led to synthetic images being used as supplementation in training datasets rather than as a primary source [20]. Given that much of the purpose of exploring synthetic images for computer vision is to use them for model training, problems with domain generalization appear to be limiting their potential.

Domain adaptation techniques are one way to address the challenge of domain generalization [35,41]. The goal of domain adaptation techniques is to enable a model trained on a source domain (where abundant labelled data is available) to perform well on a target domain (where labelled data is scarce or absent), despite differences in the data distribution between these domains. Domain adaptation techniques fall into two broad categories: data solutions, which aim to minimise the distribution gap between the source and target domains at the data level (e.g., make the domains appear more alike), and algorithmic solutions, which modify the learning algorithm to learn features that are generalizable across the domains. A relatively simple example of an algorithmic solution is maximum mean discrepancy loss (MMD loss). MMD is a statistical measure of the difference between two data distributions [42,43]. When used in a loss function, the model learns the primary task of classification while also trying to minimise the MMD distance between the source and target domain distributions [44,45]. When applied to models trained on synthetic images, domain adaptation techniques could help these models generalize to real photographs, thus improving their practicality [41].

Here we demonstrate a classification model training pipeline that predominantly uses synthetic images rendered from 3D assets to achieve predictive performance on photographs similar to a model trained entirely with real photographs, thus providing a potential solution to domain adaptation problems in synthetic biological data. Our pipeline includes three modules: 1) 3D asset creation using white-light 3D scanners, 2) rendering synthetic images in the computer graphics software "Blender", and 3) classification model training using domain adaptation techniques. The pipeline is modular and can be used in combination with other synthetic image pipelines. As a case study for our pipeline, we used the skulls from 16 terrestrial Canadian carnivore species (Order: Carnivora). As a scalable, generalizable approach that addresses the challenge of domain adaptation when using synthetic images, the pipeline we describe here demonstrates the feasibility and advantages of employing 3D assets for computer vision model training.

## Methods

In this study, we used skulls from 16 terrestrial/semi-aquatic Canadian carnivore species (Order: Carnivora), currently preserved at the Canadian Museum of Nature (Gatineau, Quebec; S1 Table, (S1 File)). Skulls served as an ideal case study for three reasons: (1) all species included in our study can be distinguished by external skull morphology, (2) the smooth surfaces of the skulls make them suitable for white-light 3D scanning, and (3) skulls are underrepresented among computer vision classification models [46]. Individual skulls were selected based on their development stage and completeness. Only adult skulls with

>50% of their teeth intact and no excessive damage were included. Mandibles were not included, primarily due to their frequent damage or omission from the remainder of the specimen, as well as the fact that they would occlude other diagnostic features on the ventral side of the skull.

All data and code is available on GitHub [47]. No permits were required for the described study, which complied with all relevant regulations.

## Data collection

**Creating 3D assets.** We scanned the skulls in batches with a Creaform GoSCAN 20 handheld scanner at an original resolution of 0.7 mm. Batches were species-specific, and batch sizes were determined by the number of skulls that could fit on a 40 cm turntable with minimum separation of ½ skull width (S1 Fig, S1 File). We scanned each batch twice (once dorsally and once ventrally) with the relative position of each skull remaining the same between scans. To keep the skulls in position for the ventral scans, we affixed them to the turntable using adhesive putty. For the dorsal scans, we simply rested the skulls in their natural position. In total, we scanned 30 skulls from each species. However, one scan of *Vulpes vulpes* (red fox, Linnaeus 1758) became corrupted and is excluded from the study. Across all the scanned specimens, there was a slight, unintentional male bias (193 males, 162 females, 124 no sex recorded). This is consistent with sex biases in mammalian natural history collections broadly [48].

After each scan was completed, we processed the 3D images using VXelements 9 [49]. First, we refined the resolution of each completed scan to 0.5 mm with the "Smart Resolution" feature of VXelements. Then we made a first pass cleaning of the image, which included removing objects such as the turntable and adhesive putty from the scan. After a pair of corresponding images (i.e., dorsal and ventral scan) had been cleaned, we used the VXelements "Merge scans" feature to merge them. We then completed a final round of cleaning in the VXmodel application of VXelements. This included filling holes in the skull (e.g., the foramen magnum) and removing artifacts of the scanning and merging process. Finally, we saved each skull image individually as a Wavefront OBJ mesh, a .mtl material file, and a bitmap texture file.

**Image collection.** To create a synthetic image dataset, we rendered 2D images of our skull assets using the open-source 3D modelling software Blender [50]. For specific Blender environment settings, refer to [47]. In Blender, we created a standard environment for each skull to be rendered in one at a time (S2 Fig, S1 File). This ensured the rendering conditions (such as lighting and camera perspective) were consistent across specimens and species, and more closely matched the real photograph imaging conditions. After a skull asset was loaded into the Blender environment, we scaled it to a standardised length (measured anterior to posterior) and decimated it to a standardised polygon count. We did this to increase rendering efficiency and ensure higher uniformity across specimens and species. Images of each skull were rendered with the 'Cycles' rendering engine from 92 angles: 18 angles (one every 20° around the yaw axis) at camera pitches of 0°, ± 30°, ± 60°, and one render at ± 90° (S3 Fig, S1 File). We automated skull loading and rendering with a Blender Python API script. Each render had a resolution of 720×720 pixels. In total, 479 3D skull assets were rendered to produce 44,068 synthetic images (Fig 1).

To create an image dataset of real photographs, we photographed an additional 10 specimens from each of the 16 species. As in the synthetic image dataset, there was an unintentional male bias among the specimens photographed (65 males, 48 females, 47 no sex recorded). None of these specimens had been included in the 3D asset dataset. As in the

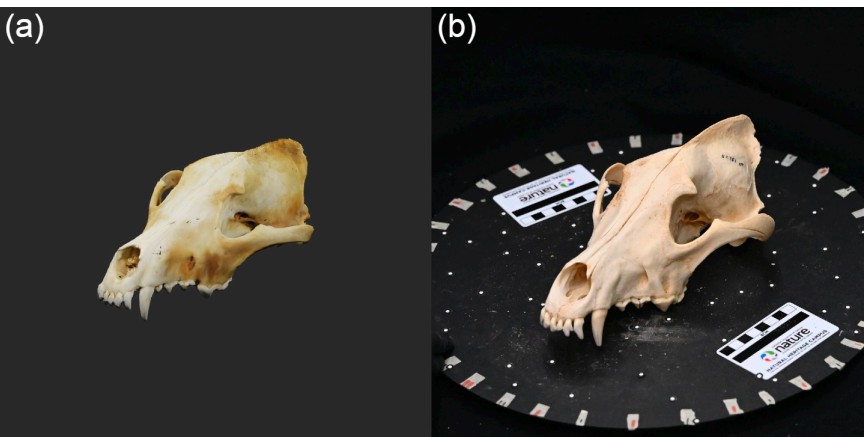

**Fig 1. Example of a rendered 3D skull image and specimen photograph.** (a) A rendered 3D skull image of a *Canis lupus* specimen. (b) A photograph of a different *C. lupus* specimen.

Blender environment, we created a standardised photography setup to minimise variation in the imaging conditions between specimens and species. However, due to the large size variation of the skulls (~8 cm to ~44 cm in length), the distance from the camera to the specimen varied depending on the species being photographed. We photographed the specimens one at a time on a black turntable in front of a black backdrop with a Laowa FFii 90mm f2.8 CA-Dreamer Macro 2X lens attached to a Nikon Z6 mirrorless camera. Each specimen was photographed from the same 92 angles as the synthetic images. Markers placed around a turntable and protractor attached to the camera tripod were used to assist with angle precision. However, due to human error in the photography process, only an average of 90.4 photos were taken per specimen. Most notably, *Neogale vison* (American mink, Schreber 1777) had no photographs taken from a camera pitch of 0°.

Using the image processing software Fiji [51], we cropped the photos to a square that would allow the skull to make a full rotation while remaining in the frame. We also removed any duplicate photos. This resulted in a total of 14,467 images for 160 specimens.

## Model training

**Data split and processing.** To create training and testing datasets, we split the sets of synthetic and photograph images separately, each at a ratio of 80:20. Splits were made at the specimen level so that all images of a single individual were only in one dataset. Despite being the target domain, we created a photograph training dataset for two reasons. The first was because the MMD model (see "Model architecture and training procedure") required unlabelled target domain images during training. The second reason was to create a baseline for comparison when a model was only trained by use of photographs. Other than in the supplemented model, the synthetic images and photographs were kept separate. This resulted in a synthetic image dataset split of 24:6 individuals and 35,328:8740 images, and a photograph dataset split of 8:2 individuals and 11,561:2906 images. The supplemented model, fine-tuned model, and subset photograph model used a 25% subset of the photograph training data, resulting in a specimen split of 2:2:6 (training:testing:unused).

We scaled all images to $224 \times 224$ pixels for training and testing, as this is the resolution required by the VGG19 architecture [52]. We applied the following augmentations to datasets that were used for supervised training: vertical and horizontal translation, horizontal flips,

and colour jitter to brightness, contrast, saturation, and hue. Testing dataset images had no augmentations.

**Model architecture and training procedure.** We built all classification models with the VGG19 feature extractor architecture with preloaded ImageNet weights [52]. To this, we added a max pooling layer, flattened layer, two dense layers, and a softmax classification layer. All models used the Adam optimizer and were trained for 100 epochs with early stopping. For all models except the supplemented model and fine-tuned model, the training stopped if there were 15 consecutive epochs with no improvements to the training domain testing loss. For example, if the model was trained by use of photographs, the early stopping criteria would be based on the photograph testing dataset loss. The supplemented and fine-tuned models had access to labelled synthetic images and photographs, so the final epoch for these models were selected manually to optimise performance across both domains. All models were trained with the PyTorch and Pytorch-Adapt libraries in Python [47,53–55].

To set baseline comparisons for how classification models performed in the absence of domain adaptation techniques, we trained three models. The first baseline model was trained by use of only the synthetic image training dataset (hereon simply referred to as 'the baseline model'). This model set a lower limit of classification performance when models were tested on photographs. The 'photograph baseline model' was trained exclusively with the photograph training dataset to set an upper-limit comparison for model performance when tested on photographs. Finally, the 'subset photograph model' was used as a point of comparison to see how a model would perform when trained with the same number of labelled photographs as the fine-tuned and supplemented model. All three baseline models were optimised by cross-entropy loss.

To improve generalization from the synthetic images to the photographs, we tested three models that each used a different domain adaptation technique. The first domain adaptation model used MMD loss to align the feature space of the synthetic images and photographs [43]. During training, the model was fed with labelled synthetic images and unlabelled photographs from each domain's respective training dataset. This implementation of MMD loss was based on [44]. The second domain adaptation model used a technique called "fine-tuning". This model was trained on the subset photograph dataset, but rather than beginning training with the ImageNet weights as in the other models, it began training from the final weights of the MMD model. Finally, we trained a 'supplemented model' which was trained with an image dataset that combined the synthetic image training dataset with the subset photograph training dataset (35,328 synthetic images and 2,907 photographs). Both the supplemented model and fine-tuned model were optimised using cross-entropy loss.

**Feature visualization.** To visualise each model's representation of the testing data's feature space, we used t-SNE on the activations of the model's post-convolution flattened layer [56]. To quantify the feature extractor's clustering ability, we measured silhouette scores from each set of t-SNE embeddings. Silhouette scores measure how well-separated clusters are in a dataset, considering both the distance within clusters (i.e., cluster tightness) and the distance between clusters (i.e., cluster separation). Clusters were defined by the ground truth labels of the images. From these clusters, we calculated three silhouette scores: by use of only synthetic images, only photographs, and with both datasets combined. Domain confusion for each model was assessed visually from the t-SNE embeddings.

To visualise which features of the images the model was using to make classifications, we created gradient-weighted class activation maps (Grad-CAMs) with 100 randomly-selected images from the photograph testing dataset [57]. The Grad-CAMs were generated by activation of the predicted class in the final convolutional layer of each model. To quantify the

types of features used by the model, we assigned each Grad-CAM a score based on the criteria in (S2 Table, S1 File). Grad-CAM scores were assigned manually and averaged for each model.

## Results

### Accuracy

When trained on exclusively synthetic images and tested on photographs, the baseline model recorded an accuracy of 55.4 (Table 1). All methods of domain adaptation produced improvements in photograph classification accuracy over the baseline model, with the supplemented model resulting in the highest classification accuracy at 95.1% (Fig 2). This was the only model to measure >90% classification accuracy on both the synthetic and photograph testing datasets (95.6% and 95.1%, respectively).

To measure how well the model would perform if it were trained exclusively on images from its own domain, we also measured accuracy on two photograph-trained models. Across all models, the photograph baseline model recorded the highest photograph classification accuracy at 99.2% (Table 1). When the photograph model was trained with the same number of labelled images as the domain-adapted models, its accuracy dropped to 85.3%, lower than both domain-adapted models that also used labelled photographs during training. When tested for domain generalization on the synthetic image test dataset, both the photograph baseline model and photograph subset model measured <30% classification accuracy.

### Qualitative analysis

**t-SNE visualisation.** The t-SNE clustering and silhouette scores showed that the feature extractors of models trained with synthetic images were better at clustering species into single, distinguishable clusters than the feature extractors of models trained only with photographs (Fig 3, Table 2). The supplemented model recorded the highest species cluster silhouette scores, regardless of whether the synthetic and photograph clusters were measured separately or together. The photograph baseline and subset models both produced poor silhouette scores for the photograph species clusters, despite their relatively high accuracy measurements when classifying photographs.

Qualitatively, overlap between domains in the t-SNE plots appeared highest in the fine-tuned model (Fig 3c). In the fine-tuned model, all individuals from a given species

**Table 1. Skull species classification accuracy for each model, as measured on the synthetic image and photograph testing datasets.** The synthetic image dataset was composed of renders of 3D skull assets and the photograph dataset was composed of photographs taken from skulls directly. The highest accuracy score for each dataset is underlined and italicised. The "Epochs" column represents the number of epochs each model was trained for. The MMD + Fine-tuning model combines the number of epochs used for the MMD model with the number of subsequent fine-tuning epochs.

| Model | Training dataset | Synthetic image accuracy | Photograph accuracy | Epochs |
|---|---|---|---|---|
| Baseline | Synthetic only | 0.952 | 0.554 | 13 |
| MMD | Synthetic + unlabelled photographs | 0.949 | 0.654 | 50 |
| MMD + Fine-tuning | MMD step: Synthetic + unlabelled photographs Fine-tuning step: Photograph subset | 0.929 | 0.896 | 50 + 2 |
| Supplemented | Synthetic + photograph subset | *0.956* | 0.951 | 24 |
| Photograph baseline | Photographs | 0.283 | *0.992* | 40 |
| Photograph subset | Photograph subset | 0.213 | 0.853 | 25 |

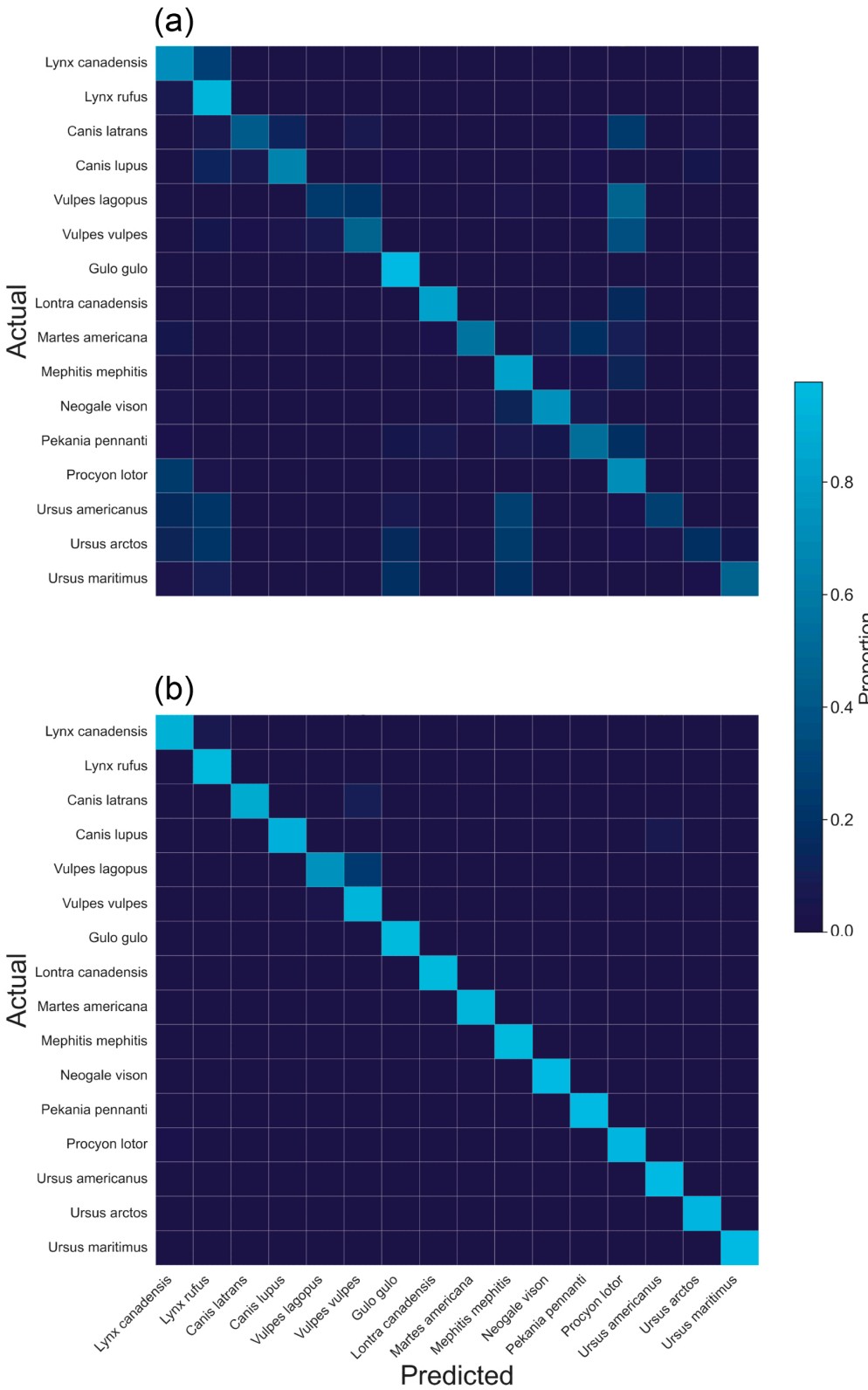

**Fig 2. Confusion matrices for the (a) baseline (synthetic only) and (b) supplemented (synthetic and photograph subset) models' classifications of the Carnivora skulls housed at the Canadian Museum of Nature.** The rows are the species' true classifications, while the columns represent the times the model made a classification as that species. Cells are shaded according to the proportion of the true labels classified as each species (i.e., shaded by row).

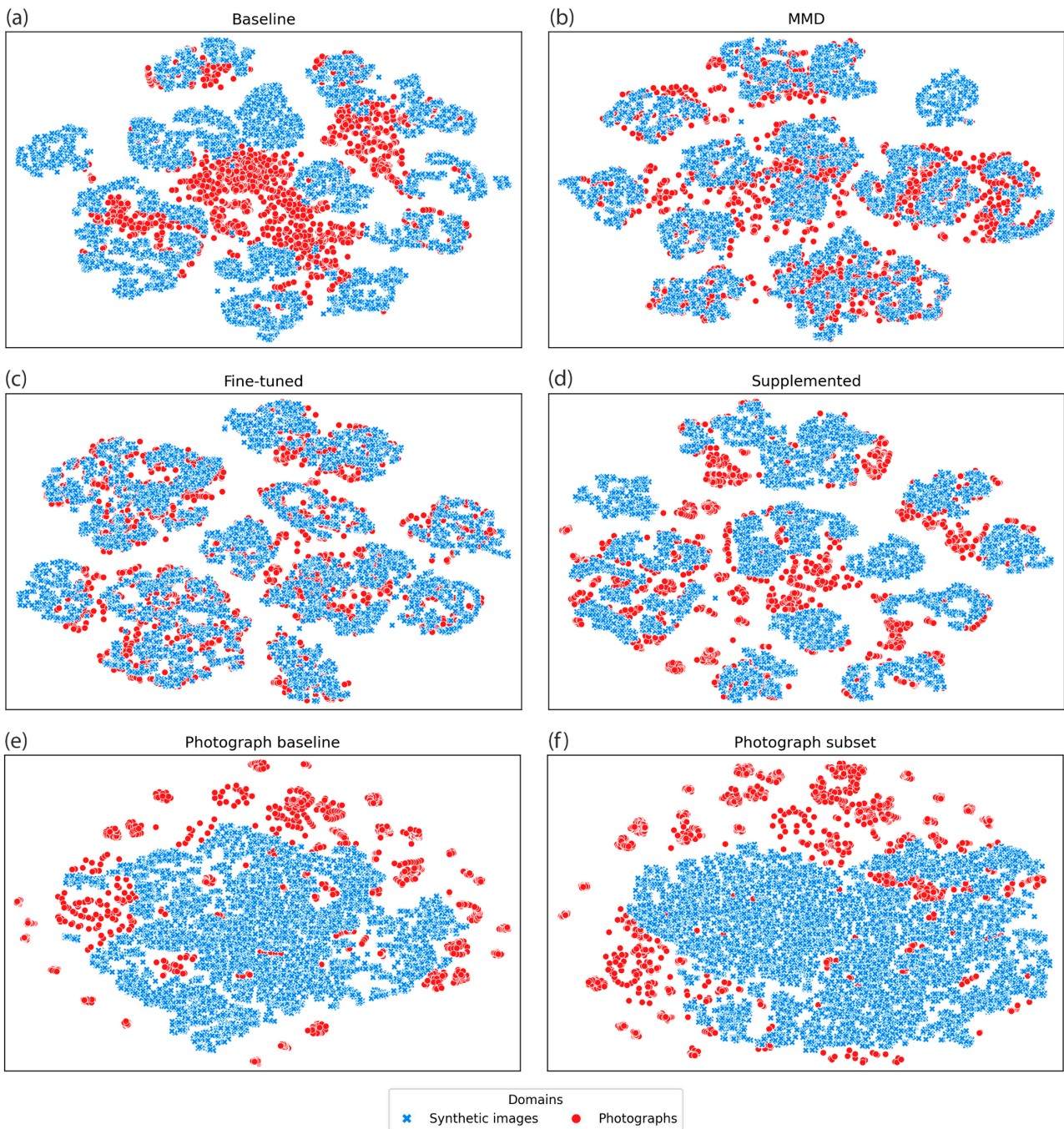

**Fig 3. Visualisation of the feature space of six skull classification models using t-SNE.** Given that the absolute axis values of t-SNE plots did not contain meaningful information, they are not shown. Each t-SNE plot was generated from using the activations of the model's post-convolution flattened layer. Blue 'x' points represent synthetic images, and the red dots represent photographs. All images were from the test dataset.

occupied the same general feature space, regardless of the image's domain. This indicated that the model was extracting features generalizable across domains, opposed to domain-specific features. The only exceptions to this were *Lontra canadensis* (North American river otter, Schreber 1777) and *Canis lupus* (grey wolf, Linnaeus 1758), which had slight shifts

**Table 2. t-SNE silhouette scores based on clusters formed from the t-SNE embeddings of each model, and labelled using the ground truth labels from each dataset.** The t-SNE embeddings were calculated from each model's activations of the final convolutional layer. The combined score measures silhouette score when the synthetic image and photograph testing datasets were combined. The highest silhouette score for each dataset is underlined and italicised.

| Model | Synthetic images | Photographs | Combined |
|---|---|---|---|
| **Baseline** | 0.447 | 0.113 | 0.297 |
| **MMD** | 0.451 | 0.116 | 0.330 |
| **MMD + Fine-tuning** | 0.353 | 0.235 | 0.325 |
| **Supplemented** | *0.462* | *0.286* | *0.378* |
| **Photograph baseline** | −0.092 | 0.016 | −0.130 |
| **Photograph subset** | −0.157 | −0.032 | −0.152 |

in feature space between domains. The MMD model had similarly high domain confusion (Fig 3b), with the key exception being the *Lo. canadensis* photographs, which clustered with *Gulo gulo* (wolverine, Linnaeus 1758) images rather than with *Lo. canadensis* synthetic images (S4 Fig, S1 File). In the supplemented model, images of the same species, but from different image domains, often formed adjacent clusters (S4 Fig, S1 File). The photograph baseline and photograph subset models showed relatively poor domain confusion (Fig 3e,f).

**Grad-CAMs.** The fine-tuned model measured the highest average Grad-CAM score, with 96 images scoring a value of 3, and four images scoring a value of 2 (Fig 4). This indicated that the model more frequently used features on the skull rather than the background to make classifications when compared to the other models. The photograph baseline and photograph subset models had the lowest average Grad-CAM scores, as both models had 50 or more images with a score of 0.

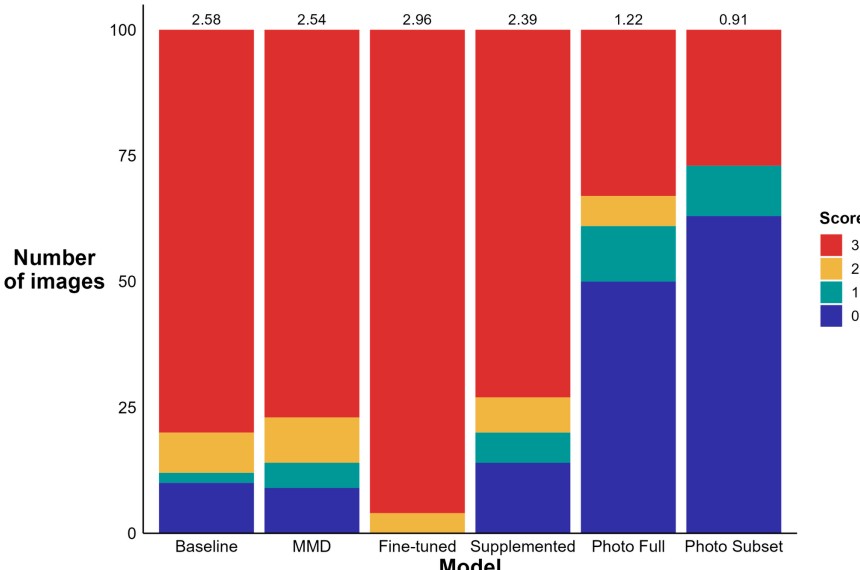

**Fig 4. Grad-CAM scores for each model.** The average scores for each model are printed at the top of the model's bar. High Grad-CAM scores (i.e., closer to 3) indicated that the model frequently focussed on skull morphology to make classifications opposed to background features. Exact Grad-CAM scoring criteria can be found in (S2 Table, S1 File).

## Discussion

### The pipeline

Domain generalization is a challenge for using 3D assets as a source of training images in classification models for biodiversity monitoring [20,41]. In this study, we present a simple, effective, and modular pipeline to train domain-generalizable classification models primarily using synthetic images generated from 3D assets. Through the pipeline's three components (3D scanning, image rendering, and model training), the pipeline explicitly addresses classification generalizability on two fronts. First, for synthetic image generation we created a Blender environment that automatically produced 92 multiview images of each skull. The Blender environment also allowed for a high degree of precision in image angles, thus ensuring maximum diversity in the angles each skull was imaged from. The limit to the number of images produced for each skull was also entirely self-imposed, and the Blender environment can be modified to effectively have no limit on the number of unique images produced for each skull. Second, by using domain adaptation training procedures such as MMD loss, we were able to produce models that were highly accurate when tested on both source domain and target domain images (synthetic images and photographs, respectively).

The simplicity and modularity of this pipeline allow it to potentially be applied to a wide range of taxa and objectives. The pipeline's three primary components all function independently of each other, and thus can be substituted with alternative methods without affecting the other components. For example, the 3D assets produced via scanning could be subjected to more complicated rendering methods that allow the assets to be posed and rendered more realistically, such as in replicAnt [22], which uses a video game graphics engine to render 3D assets in realistic environments for use in computer vision models. Scanning can also be swapped out with other 3D asset creation methods such as photogrammetry, which have the advantage of potentially being more affordable than 3D scanners [21,28,29]. While the subjects of the classification models can obviously be substituted with other taxa, the models themselves are not rigid either and can be customized to fit various classification problems (changing the model architecture, using alternative domain adaptation techniques, etc.).

### Domain adaptation

Here we show that domain adaptation methods such as MMD loss and fine-tuning can significantly improve classification performance on target domain images (Table 1). On its own, MMD loss improved classification performance over the baseline model, but still underperformed both models trained exclusively with real photographs. Accuracy of the supplemented and fine-tuned models surpassed the photograph subset model (+9.8% and +4.1% respectively), even though all three models were exposed to the same number of labelled photographs during training. This shows that in the absence of a large, labelled target domain dataset, training datasets primarily composed of synthetic images can yield high classification accuracy.

Even in cases where large, labelled target domain datasets are available, the inclusion of synthetic images during training might still warrant consideration. A known challenge in ecological applications of computer vision is that models can learn contextual clues in images (e.g., background features) to inform classification decisions [58]. In moderation this might be acceptable, as the inclusion of relevant contextual metadata (e.g., spatiotemporal data) were shown to improve model performance [59,60]. However, if the contextual clues are not generalizable to new situations, such as the markings on the turntable in our photograph

dataset, or if the contextual data becomes more important to the model than the morphological features of the specimen, this becomes problematic. As we have shown by testing our photograph models on synthetic images (Table 1), models that rely heavily on contextual features generalize poorly to new situations. An advantage of synthetic images is that all features of the images can be tightly controlled by the dataset's creator. In the synthetic images used for this study, we ensured that all contextual features of the images were uniform across species, thus forcing the model to exclusively learn from the morphology of the specimens. This yielded a more generalizable baseline model (55.4% cross-domain accuracy vs. 28.3%), which was further enhanced using domain adaptation techniques such as MMD loss and fine-tuning. Here we have shown that by first training a model on a dataset with generalizable features (such as our synthetic image dataset), and then adapting that model to the real-life domain in which it will be deployed, a model can be encouraged to learn relevant, generalizable features while still maintaining high classification accuracy (Fig 4).

## Conclusion

Collecting labelled images is an obstacle to building computer vision models for ecological applications due to the high quantity of images required and the human resources needed to collect and label said images. Leveraging the wealth of readily available biological samples in natural history collections to create 3D models for synthetic images generation is an interesting solution to this problem, but so far has faced its own challenge of domain adaptation. In this study, we report a simple approach to producing generalizable classification models trained primarily by use of 3D assets generated from museum specimens. Our work is a step towards unlocking the full potential of synthetic images and museum collections in the context of computer vision for ecology.

## Supporting information

**S1 File. Supporting figures and tables.**
(PDF)

## Acknowledgments

We thank the Canadian Museum of Nature for allowing access to their specimens, equipment, and facilities. We thank Greg Rand, Marie-Helene Hubert, and Alan McDonald for their assistance with the specimen collections and scanning equipment. We also thank Drs Leonid Sigal, Michelle Tseng, and Rachel Germain for their insightful discussions.

## Author contributions

**Conceptualization:** Kamal Khidas, Katie E. Marshall.

**Data curation:** Jarrett D Blair.

**Formal analysis:** Jarrett D Blair.

**Funding acquisition:** Jarrett D Blair, Kamal Khidas, Katie E. Marshall.

**Investigation:** Jarrett D Blair.

**Methodology:** Jarrett D Blair.

**Project administration:** Jarrett D Blair, Kamal Khidas, Katie E. Marshall.

**Resources:** Kamal Khidas, Katie E. Marshall.

**Software:** Jarrett D Blair.

**Supervision:** Kamal Khidas, Katie E. Marshall.

**Validation:** Jarrett D Blair, Kamal Khidas, Katie E. Marshall.

**Visualization:** Jarrett D Blair.

**Writing – original draft:** Jarrett D Blair.

**Writing – review & editing:** Jarrett D Blair, Kamal Khidas, Katie E. Marshall.

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
