## [Decision Letter · Decision Letter 0]

2 May 2025

PONE-D-25-15313Leveraging synthetic data produced from museum specimens to train adaptable species classification modelsPLOS ONE

Dear Dr. Blair,

Thank you for submitting your manuscript to PLOS ONE. After careful consideration, we feel that it has merit but does not fully meet PLOS ONE’s publication criteria as it currently stands. Therefore, we invite you to submit a revised version of the manuscript that addresses the points raised during the review process.

We look forward to receiving your revised manuscript.

Kind regards,

Gianniantonio Domina, Ph.D.

Academic Editor

PLOS ONE

Journal Requirements:

3. In your manuscript, please provide additional information regarding the specimens used in your study. Ensure that you have reported human remain specimen numbers and complete repository information, including museum name and geographic location.

For more information on PLOS ONE's requirements for paleontology and archeology research, see https://journals.plos.org/plosone/s/submission-guidelines#loc-paleontology-and-archaeology-research.

4. Please note that PLOS ONE has specific guidelines on code sharing for submissions in which author-generated code underpins the findings in the manuscript. In these cases, we expect all author-generated code to be made available without restrictions upon publication of the work. Please review our guidelines at https://journals.plos.org/plosone/s/materials-and-software-sharing#loc-sharing-code and ensure that your code is shared in a way that follows best practice and facilitates reproducibility and reuse.

“This work was supported by a Mitacs Elevate grant to J.D.B, as well as contributions from the Canadian Museum of Nature Foundation. We thank the Canadian Museum of Nature for allowing access to their specimens, equipment, and facilities. We thank Greg Rand, Marie Helene Hubert, and Alan McDonald for their assistance with the specimen collections and scanning equipment. We also thank Drs Leonid Sigal, Michelle Tseng, and Rachel Germain for their insightful discussions.”

“JDB received a Mitacs Accelerate grant to fund this work. The grant does not have a specific grant number. Mitacs' website can be found at https://www.mitacs.ca/. The funders did not play any role in the study design, data collection, decision to publish, or preparation of the manuscript.”

**Additional Editor Comments:**

This MS deserves to be published on PLOS One. It is well written in a scientific form. The authors have to follow the suggestions of the reviewer to prepare your imporved MS. In addition do not have to forget to add the authors of species names the first time that them are cited.

Reviewers' comments:

Reviewer's Responses to Questions

**Comments to the Author**

1. Is the manuscript technically sound, and do the data support the conclusions?

Reviewer #1: Yes

2. Has the statistical analysis been performed appropriately and rigorously? 

Reviewer #1: Yes

3. Have the authors made all data underlying the findings in their manuscript fully available?

Reviewer #1: Yes

4. Is the manuscript presented in an intelligible fashion and written in standard English?

Reviewer #1: Yes

5. Review Comments to the Author

Reviewer #1: Dear authors, I find the manuscript well written. It seems to me that the strategy you propose is solid and could be applied to several domains other than the example you provide.

I have one major concern only. In line 201, you state that you scaled the images to 224x224 pixels for both training and testing. It seems to me that this low resolution could bias the performance of the model, since it could lead to loss of details in the images. Can you please explain why you decided to operate at this resolution, and why? Have you compared other resolutions, in order to decide the one with the best cost/effectiveness ratio? Please, provide some metrics.

Best regards

SM

6. PLOS authors have the option to publish the peer review history of their article (what does this mean?). If published, this will include your full peer review and any attached files.

Reviewer #1: **Yes: **Stefano Martellos

---

## [Author Response · Author response to Decision Letter 1]

23 Jun 2025

Please see the uploaded response to reviewers document.

---

## [Editor Report · Decision Letter 1]

17 Jul 2025

Leveraging synthetic data produced from museum specimens to train adaptable species classification models

PONE-D-25-15313R1

Dear Dr. Blair,

We’re pleased to inform you that your manuscript has been judged scientifically suitable for publication and will be formally accepted for publication once it meets all outstanding technical requirements.

Kind regards,

Gianniantonio Domina, Ph.D.

Academic Editor

PLOS ONE
---

## [Editor Report · Acceptance letter]

PONE-D-25-15313R1

PLOS ONE

Dear Dr. Blair,

I'm pleased to inform you that your manuscript has been deemed suitable for publication in PLOS ONE. Congratulations! Your manuscript is now being handed over to our production team.

Kind regards,

on behalf of

Prof. Gianniantonio Domina

Academic Editor

PLOS ONE